



# Mahalanobis distance based recognition of changes in the dynamics of seismic process

Teimuraz Matcharashvili[1], Zbigniew Czechowski[2], Natalia Zhukova[1]

1. M. Nodia Institute of Geophysics, Tbilisi State University, Tbilisi, Georgia
2. Institute of Geophysics, Polish Academy of Sciences, Warsaw, Poland

## Abstract

In present work we aimed to analyze regularity of seismic process based on all its spatial, temporal and energetic characteristics. Increments of cumulative times, increments of cumulative distances and increments of cumulative seismic energies, have been calculated from southern California earthquake catalogue, 1975 to 2017.

As the method of analysis we used multivariate Mahalanobis distance calculation which was combined with the surrogate data testing procedure - often used for testing of nonlinear structure in complex data sets. Prior to proceed to the analysis of dynamical features of seismic process we have tested used approach for two different 3 dimensional models in which dynamical features were changed from more regular to the more randomized conditions by adding some extent of noises.

Analysis of variability in the extent of regularity of seismic process have been accomplished for different representative threshold values.

According to results of our analysis about third part of considered 50 data windows, the original seismic process is indistinguishable from random process by its features of temporal, spatial and energetic variability. It was shown that prior to strong earthquake occurrences, in periods of relatively small earthquakes generation, percentage of windows in which seismic process is indistinguishable from random process essentially increases (to 60-80%). At the same time, in periods of aftershock activity in all considered windows the process of small earthquake generation become regular and thus is strongly different from randomized catalogues.

In some periods of catalogue time span, seismic process looks closer to randomness while in other cases it becomes closer to regular behavior. Exactly, in periods of relatively decreased earthquake generation activity (at smaller energy release), seismic process looks random-like while in periods of occurrence of strong events, followed by series of aftershocks, it reveal significant deviation from randomness - the extent of regularity essentially increases. The period, for which such deviation from the random behavior can last, depends on the amount of seismic energy released by the strong earthquake.

## Introduction

The process of earthquakes generation still remains in the focus of diverse interdisciplinary investigations of Earth science researchers worldwide. Practical and scientific reasons for such interest are well known and easily explainable. At the same time, despite of great interests and already applied enormous research efforts, currently many important aspects of the complex seismic process characterized by the space and time clustering are still not clear [Bowman&Sammis,2004; Godano&Tramelli, 2016; Matcharashvili et al. 2018; Pasten et al. 2018].

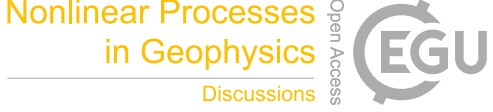

One of such fundamental questions of nowadays Earth sciences concerns dynamics of seismic process. As a logical compromise, between different possibilities proposed on this problem, it has been suggested that dynamical features of seismic process may be diverse and range from periodic (mostly for large events) to totally random occurrence of earthquakes [Matcharashvili et al. 2000; Corral, 2004; Davidsen&Goltz, 2004].

The same, in terms of earthquakes generation intermittent criticality concept, can be expressed as an ability of tectonic system to approach and/or retreat from a critical state - state of system in which strong earthquakes occur [see e.g. Sornette&Sammis, 1995; Bowman, et al, 1998; Bowman&Sammis, 2004; Corral, 2004].

Current knowledges about scaling and memory characteristics of the whole seismic process indeed supports mentioned above diversity of dynamics of earthquakes generation [Sornette&Sammis, 1995; Bowman, et al, 1998; Suzuki, 2004; Chelidze and Matcharashvili, 2007; Czechowski, 2001, 2003, Białecki and Czechowski 2010]. Moreover, results of analysis carried out to assess dynamical features of seismic process in its separate domains (time, space and energy) also indicates different behavior [see e.g. Goltz,1998; Matcharashvili et al., 2000, 2002; Abe and Suzuki, 2004; Chelidze and Matcharashvili, 2007; Iliopoulos et al.,2012]. Exactly, it was shown, that seismic process in the temporal and spatial domains may reveal features which are close to so called low-dimensional dynamical structure, though by features of behavior in the energy domain it looks like close to randomness i.e. represent high-dimensional dynamical process [Goltz, 1998; Matcharashvili et al.,2000; Iliopoulos, et al. 2012]. This was shown for whole catalogues as well as for its spatial parts or for different time periods.

Coming back to the concept of critical state it needs to be underlined that intermittent criticality implies time-dependent variations in the activity during a seismic cycle. So, as far as critical state usually is described as the state of the system when it is at the boundary between order and disorder [Bowman et al. 1998] we should describe time variability of seismic process in terms of order or disorder. In this respect it is crucially important to point what is meant under the term order (or disorder) in this sense. In common parlance it looks intuitively understandable that when someone is facing a strong destructive event, after a seismically calm period (with small earthquakes), it may really seem that the order has been replaced by disorder. At the same time, the nature of such a transition should be strictly described in terms of contemporary concept of geocomplexity [Rundle, et al. 2000].

According to present knowledges, in complete accordance with the intermittent criticality concept, it is accepted that the extent of regularity (order) of the seismic process may vary in all its domains (temporal, spatial and energetic)[Goltz, 1998; Abe and Suzuki, 2004; Chelidze and Matcharashvili, 2007; Iliopoulos et al., 2012; Matcharashvili et al., 2000, 2002, 2018]. At the same time, despite the large enough number of recent publications evidencing the diversity of such changes in the dynamics of the seismic process, interest to the question still continues to grow. In this regard, it needs to be emphasized the importance of assessing of dynamical changes on the basis of multivariate analysis, taking into account all the temporal, spatial and energetic constituents of the seismic process. Thus, the important research task is to understand character of such changes of entire seismic process.

Based on all above mentioned, in present work we aimed to investigate dynamical features of seismic process based on all its temporal, spatial and energetic characteristics. Namely, we accomplished multivariate comparison of seismic process from original south Californian earthquake catalogue and from the set of randomized catalogues in which unique (temporal, spatial and energetic) dynamical structures have been intentionally distorted by shuffling procedure. This

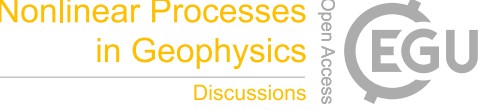



enabled to assess where and how dynamics of original seismic process is close to disorder
(irregularity) or to order (regularity).
It was shown that extent of regularity in seismic process is close to randomness in periods
prior to strong earthquakes. After strong earthquakes, the regularity of original seismic process
assessed by used temporal spatial and energetic characteristics is clearly increased.
98                              **Used data and Methods of analysis**
We base our analysis on the southern California earthquake catalog available from
http://www.isc.ac.uk/iscbulletin/search/catalogue/). We focused on the time period from 1975 to
2017 (see Fig. 1). According to results of time completeness  analysis the southern California (SC)
earthquake catalog for the considered period is complete for M=2.6.

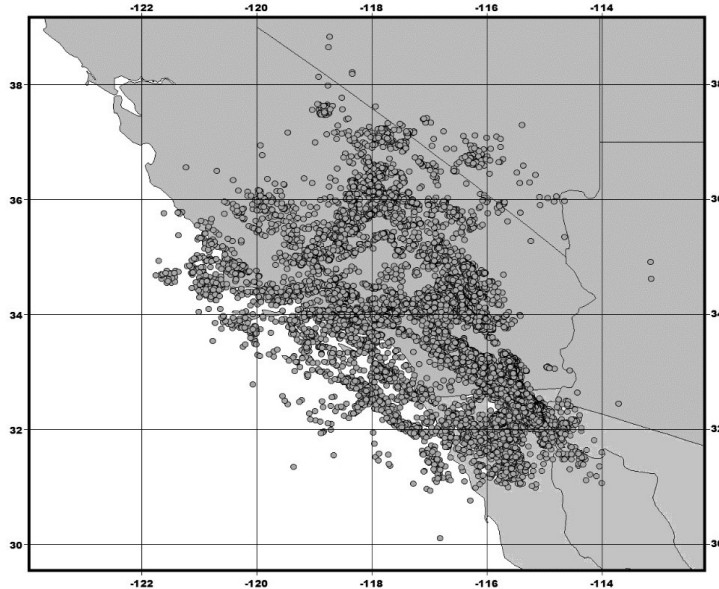

Fig. 1. Map of area covered by southern California (SC) earthquake catalog (1975-2017).
As we pointed above, we aimed at the multivariate analysis of dynamical features of
seismic process. Therefore, in order to preserve original character of temporal, spatial and
energetic characteristics of considered process we intentionally avoided any cleaning or filtering
of used earthquake catalogue. Here we are based on a common and already accepted practice [see
e.g. Bak et al. 2002; Christensen et al. 2002; Corral, 2004; Davidsen&Goltz, 2004; Matcharashvili
et al. 2018]; namely we putted all events on the same footing and considered catalogue as a whole.
In other words, we do not paid attention to the details of tectonic features, earthquakes location or
their classification as mainshocks or aftershocks [Bak et al. 2002; Christensen et al. 2002; Corral,
2004]. For further clarity we declare that take responsibility on the trustworthy of our analysis,
assuming meanwhile that used SC catalogue is a result of careful work of skilled professionals and





thus represents reliable collection of necessary for our study data (in other words we take
responsibility according to third point listed in Madigan et al. [2014]).

Thus, we aimed to accomplish the multivariate assessment of changes in the extent of
regularity of the original seismic process. According to this research goal, we needed to analyze
seismic process in the terms of the simultaneous variability in all three its domains – temporal,
spatial and energetic. From this point of view we consider cumulative sums of earthquakes
characteristics in temporal, spatial and energetic domains (Fig.2). The cumulative sum
representation in the time domain is trivial as far as time is already cumulative characteristic
representing cumulative sum of inter-earthquakes times. Cumulative representation in spatial
domain is also quite feasible and there is not any logical problems against consideration of
cumulative sums of distances between consecutive earthquakes in seismic catalogue. As for
cumulative sum of seismic energies, released by consecutive earthquakes, this characteristic is
often used in the context of different aspects of earthquake generation [e.g. Bowman, 1998, 2008;
Nakamichi et al 2018]. Here we add that despite of some controversies [for references see Corral,
2004, 2008] in the question of reliable energetic measurement of earthquake size, anyway its
proportionality with the earthquake magnitude is generally accepted. Thus, from SC catalogue
earthquake magnitudes we calculated amount of released seismic energy according to Kanamori,
[1977].

Hence, beginning from the starting (first) earthquake in the considered catalogue, we can
characterize each of consecutive earthquakes in terms of corresponding increments of cumulative
time - *ICT(i)*, increments of cumulative distances - *ICD(i)* and increments of cumulative seismic
energies - *ICE(i)*. Each of these data sets, of derivative quantities *(ICT(i), ICD(i), ICE(i))*, has been
normed to its' standard deviation.

Next we needed to choose appropriate to research goal method of analysis by which we
could characterize seismic process from multivariate point of view. For this we used the well
known statistical test of Mahalanobis distance (MD) calculation. MD calculation is effective
multivariate method for different classification purposes and is often used for data sets of different
origin. Thus, the objective of our analysis can be regarded as a classification task, having in mind
the features of seismic process assessed by the variability of *ICT(i), ICD(i)* and *ICE(i)*
characteristics.

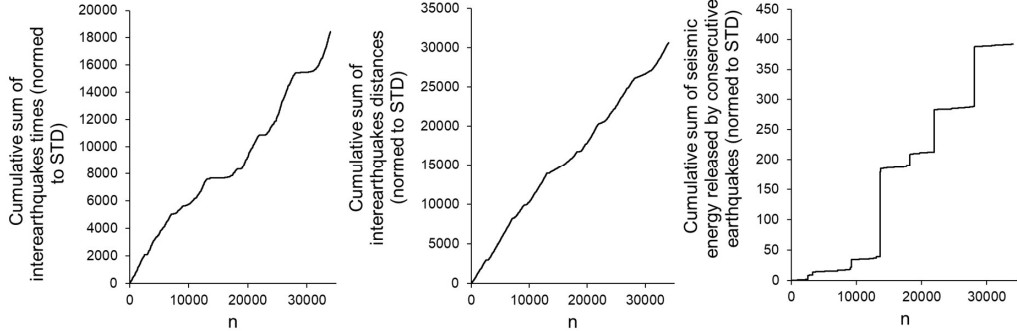

Fig. 2. Cumulative sums of interevent times (a), inter-earthquake distances (b) and released seismic
energies (c), starting from the first event in SC catalogue (1975-2017).

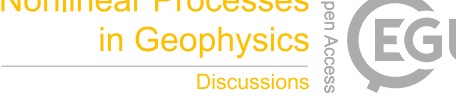



In other words, we aimed to assess changes that occurred in the seismic process for the
period of southern Californian catalogue span (1975-2017). Generally it is well known that
correctness of systems' multivariate assessment and classification is strongly depended on the
correct feature extraction [McLachlan, 1992, 1999]. To be more precise it need to be added that,
it is important that used data sets are to be exactly focused on targeted features of the investigated
process. For this, in order to have data sets of similar physical sense enabling to assess dynamical
features of seismicity, as was mentioned above, we selected *ICT(i), ICD(i)* and *ICE(i)* data sets.
Next we needed to derive a quantitative measure for reliable comparison of the seismic process
based on these characteristics.
Usually, comparing groups of discriminant variables, one compare the centroids for these
groups, instead to compare just the mean values of variables. In this way, in terms of multiple (in
our case three) characteristics, we will get a measure of the divergence or the distance between the
compared groups. This gives opportunity to make conclusion on the question whether  investigated
groups are similar or dissimilar by targeted characteristics. As we pointed above, for such purposes
we used method of MD calculation [Mahalanobis, 1930; McLachlan, 1992, 1999]. A Mahalanobis
distance (often denoted also as *D*) can be calculated from the following expression (1):

$$D^2 = (\bar{x}_1 - \bar{x}_2)^T S^{-1} (\bar{x}_1 - \bar{x}_2) \qquad (1)$$

where $\bar{x}_1$ and $\bar{x}_2$ are mean vectors of sample sets (of *ICT(i), ICD(i)* and *ICE(i)* data from original
and randomized catalogues) of sizes $n_1$ and $n_2$, and *'T'* superscript denotes the transpose operator.
*S* is the pooled covariance matrix:

$$S = \frac{((n_1-1)S_1 + (n_2-1)S_2)}{n_1+n_2-2} \qquad (2)$$

where  $S_i$ are the covariance matrices of the corresponding groups.
Generally, two conditions or states of systems are more probable to fall in the same class
or group (or are similar at higher probability) in the case when calculated MD value is smaller. In
order to assess the significance of the difference between the groups, the Hotelings $T^2$ statistics
was used, converted into an *F*–value and assessed by an *F*-test.  Exactly, the *F* value was calculated
as:

$$F = \frac{n_1 n_2}{n_1+n_2} \frac{n_1+n_2-p-1}{(n_1+n_2-2)p} D^2 \qquad (3).$$

In (3)  *p* is the degrees of freedom. After in order to make final conclusion about the similarity or
dissimilarity of analysed groups we compared calculated *F* values with a critical value, $F_c$
(corresponding to the degrees of freedom).  In case if $F>F_c$, the statistically significant difference
between the groups is established, at a specific probability (significance level).
Dealing with analysis of complex seismic process it should be pointed that the MD
calculation is sensitive to inter-variable changes in a multivariate system [Mahalanobis, 1930;
Lattin et al. 2003] and that it takes into account the correlation among several variables providing
information about similarity or dissimilarity between compared groups [Taguchi &Jugulum, 2002;
Kumar et al. 2012].
As far as most interesting is to analyze dynamical changes occurred on short scales (short
data sets) it is useful to combine advantages of multivariate analysis and surrogate testing
[Matcharashvili 2017, 2018]. Exactly, we can use multivariate Mahalanobis distance calculation
to see whether original seismic process is similar or is dissimilar with the random  process
(randomized catalogues), comparing them by listed above three main characteristics.





As mentioned we aimed to analyze how the extent of order in the seismic process, assessed
by its derivative temporal, spatial and energetic characteristics (quantities of *ICT(i), ICD(i)* and
*ICE(i)*), is changing over the period of analysis. For this we compared the original catalogue, with
the set of artificial catalogues in which the original dynamical structures (of temporal, spatial and
energetic distributions) have been intentionally destroyed by the shuffling procedure
[Kantz&Schreiber, 1998]. We have generated 100 of such randomized catalogs.
In order to test whether the used approach, combining MD calculation and surrogate
testing, may indeed be useful to discern changes that may occur in the natural 3D system (seismic
process in tectonic system), with slightly or strongly different dynamical features, we used series
of simulated 3 dimensional systems with added noises. Namely, 3D Lorenz system and crack
fusion model with added Gaussian noises.
**Lorenz model.** The well known Lorenz model describes the motion of an incompressible fluid
contained in a cell that have a higher temperature at the bottom and a lower temperature at the top.
In spite of its simple form of the set of equations it can exhibit very complex behaviors. Therefore,
it has been commonly used to presentation of an interesting nonlinear dynamics of 3D systems.
The Lorenz model has the following form [see e.g., Hilborn, 1994]:
$$\frac{dx}{dt} = p(y - x)$$
$$\frac{dy}{dt} = -xz + rx - y \qquad (4)$$
$$\frac{dz}{dt} = xy - bz$$

where $p$ represents the Prandtl numer, $r$ – the Reyleigh numer and $b$ is related to the ratio of the
vertical height of the fluid layer to the horizontal size of the convection rolls. For  parameter $r < 1$
trajectories in 3D space $(x, y, z)$ are attracted by the origin $(0, 0, 0)$. When $r > 0$ the Lorenz model
has three fixed points which can have different features.
In this work we need stationary-like time series, therefore in order to avoid periodic orbits we
assume $r < 1$, namely $r = 0.7$.  In order to generate time series we use the discrete version of the
Lorenz equations modified by introducing two random noises:
$$x_{t+\Delta t} = p(y_t - x_t)\Delta t + x_t + c\xi_t + \varepsilon\zeta_x$$
$$y_{t+\Delta t} = (-x_t z_t + rx_t - y_t)\Delta t + y_t + c\xi_t + \varepsilon\zeta_y \qquad (5)$$
$$z_{t+\Delta t} = (x_t y_t - bz_t)\Delta t + z_t + c\xi_t + \varepsilon\zeta_z$$

First noise, $\xi$, is the same (i.e., has the same values) in the three equations and for all cases under
investigation. Its role is keeping states of the system around the attractor in the origin $(0, 0, 0)$. The
Lorenz model with noise $\xi$ only, will be treated as a basic reference ('deterministic') system. The
second noise $\zeta_x$ ($\zeta_y$ and $\zeta_z$) will be generated separately for each of the three equations. It is
multiplied by the parameter $\varepsilon$ with increasing values. The role of the second noise is checking the
influence of increasing randomness on the measures of order in the process.  For generation of
time series by the system (5) we assume the following values for parameters, $p = 10$, $r = 0.7$, $b =$





8/3, $c = 3$, the initial values $(x(0), y(0), z(0)) = (0, 0, 20)$, and the time step $\Delta t = 0.001$. The
parameter $\varepsilon$ will increase from 0.0 (for the reference system) to 1.0.
**Crack fusion model.** The kinetic crack fusion model [Czechowski, 1991, 1993, 1995] describes
the evolution of a system of numerous cracks which can nucleate, propagate and coalesce under
the applied stress. Here we use a simply version of the model (related to seismic processes) where
only three crack populations (small cracks $x(t)$, medium cracks $y(t)$ and big cracks $z(t)$) are taken
into account. Their evolution is governed by the following system of nonlinear equations:

$$\frac{dx}{dt} = -a(1-k_x)xx - axy - axz + bz + \mu T$$

$$\frac{dy}{dt} = a(k_y - k_x)xx - a(1-k_y)yy - a(1-2k_y)xy - ayz \qquad (6)$$

$$\frac{dz}{dt} = \frac{1}{2}a(1-2k_y)(xx + 2xy + yy) - \frac{1}{2}azz - gz$$

where parameters $a, k_x, k_y$ are related to the coalescence probability, $b$ is a nucleation rate of small
cracks around big cracks, $g$ is a healing rate of big cracks. The second source term for small cracks
is due to the external stress $T(t)$ which can grow in response to relative tectonic plate motion and
can diminish according to the number of big cracks $z(t)$, i.e.
$$\frac{dT}{dt} = \begin{cases} v(1-z), & T \geq 0 \\ 0, & T < 0 \end{cases} \qquad (7)$$

Similarly as the Lorenz model, the crack fusion model exhibits two kinds of behavior: it can decay
to the one stationary point or its attractor can be given by periodic orbits. Because (like before) we
need stationary-like time series, so in order to avoid periodic orbits we assume the parameters: $v\mu$
$= 1000 < (v\mu)_{crit} = 6320$ and modify the hierarchical system by introducing two random noises: $\xi$
and $\zeta_x$ to the equation for small cracks only.
$$x_{t+\Delta t} = (-a(1-k_x)x_t x_t - ax_t y_t - ax_t z_t + bz_t + \mu T_t)\Delta t + x_t + c\xi_t + \varepsilon\zeta_x$$

$$y_{t+\Delta t} = (a(k_y - k_x)x_t x_t - a(1-k_y)y_t y_t - a(1-2k_y)x_t y_t - ay_t z_t)\Delta t + y_t \qquad (8)$$

$$z_{t+\Delta t} = \left(\frac{1}{2}a(1-2k_y)(x_t x_t + 2x_t y_t + y_t y_t) - \frac{1}{2}az_t z_t - gz_t\right)\Delta t + z_t$$

In order to generate time series by the system (8) we assume the following values for parameters,
$a = 8, b = 20, c = 0.5, g = 1, k_x = 0.3, k_y = 0.45, v = 10, \mu = 100$, the initial values $(x(0), y(0), z(0))$
$= (0, 0, 20)$, and the time step $\Delta t = 0.01$. The parameter $\varepsilon$ will increase from 0.0 (for the reference
system) to 0.35.
**Results and discussion**
In Fig. 3, we present results of MD calculation for non-overlapping 50 data windows
shifted by 50 data. Here are compared 50 data groups each of which contained columns of *ICT(i)*,

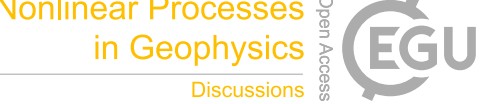

*ICD(i)* and *ICE(i)* sequences. Exactly, groups consisting of *ICT(i)*, *ICD(i)* and *ICE(i)* columns
from original catalogue were compared with groups of corresponding three columns consisted of
averaged for 100 randomized catalogues, *ICT(i), ICD(i), ICE(i)* data.

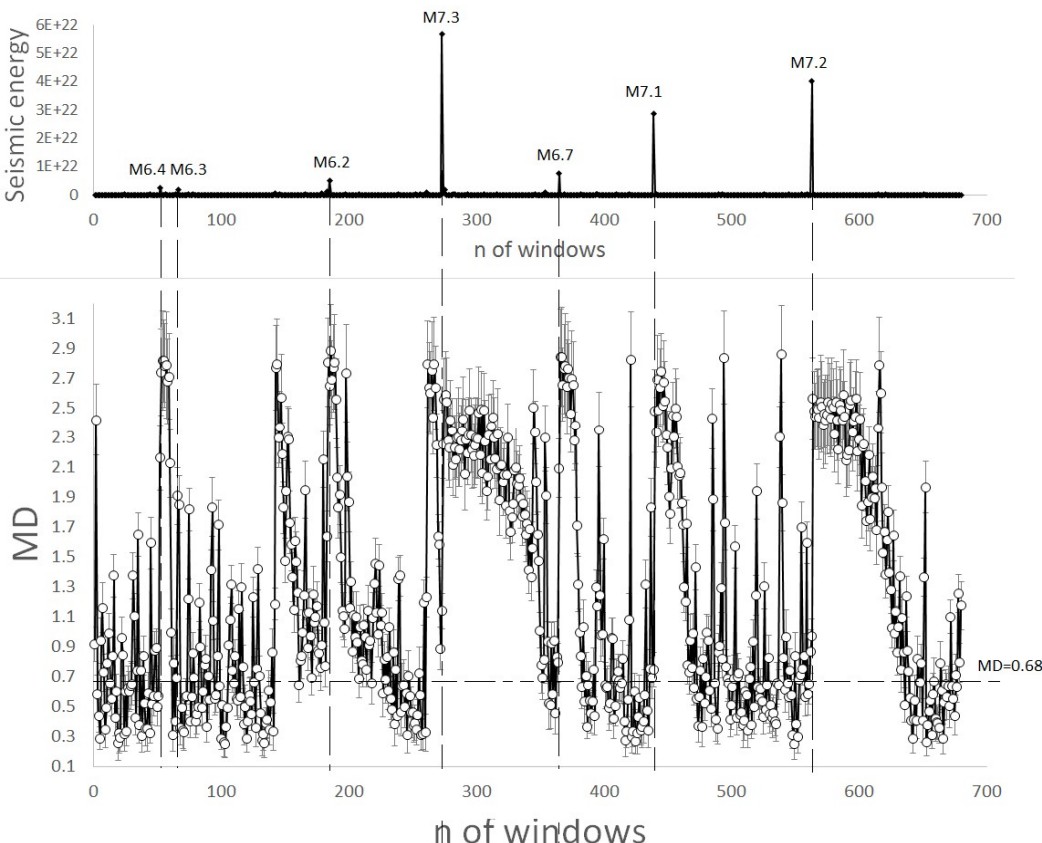


Fig. 3. Released seismic energy (top curve) and averaged MD values (bottom curve) calculated for
consecutive non-overlapping 50 data windows, shifted by 50 data, in Southern California earthquake
catalogue (1975-2017). MD values were calculated by comparing *ICT(i), ICD(i)* and *ICE(i)* sequences in
the original catalogue and in the set of randomized catalogues. Dotted line corresponds to significant
difference between windows at p=0.05.

In order to be further convinced, that the used multivariate method enables to discriminate
different conditions of dynamical systems, as is mentioned above we decided to use 3 dimensional
models in which dynamical features were changed from more regular to the more randomized
conditions by adding some extent of noises. We started from the Lorenz system (Fig. 4) and then
proceeded to crack fusion model [Czechowski, 1991, 1993, 1995] (Fig.5). As it is said in previous
section, in both cases to original 3D system we additionally added noise of different intensity
assuming that as more intense is added noise the closer to randomness should be analyzed model
system. In figures below (Fig. 4 and 5) it is clearly shown that the number (or portion) of 50 data





windows in which condition of 3D system is indistinguishable from the initial condition (system
with no added noise) gradually decreases when the intensity of added noise increases. This means
that used method of analysis enables to distinguish conditions of systems even in cases when they
are just slightly different (only small amount of noise is added) (see left parts of curves in Figs. 4
and 5, at smaller amount of added noise intensity).
For clearness we add here that in Figs. 4 and 5, we focused on the case of 50 data long
windows because in the further analysis we also used 50 data windows for the seismic catalogue
analysis. At the same time, it should be underlined that the result of above analysis is depending
on the used time scale (size of windows). In case of larger windows (500 data 1000 data etc.)
distinguishability from the starting condition (without added noise) necessitates larger amount of
added noise, though general conclusion remain the same – used method of analysis enables to
distinguish conditions of 3D systems with different extent of dynamical regularity.

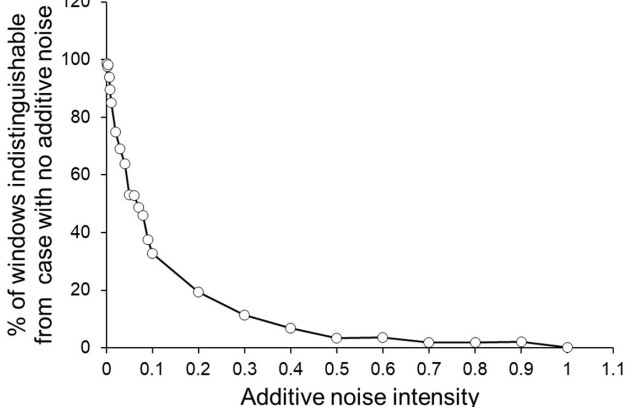


Fig. 4. Percentage of 50 data windows, shifted by 50 data step, of Lorenz system with added noise
indistinguishable from the initial condition (system with no added noise).

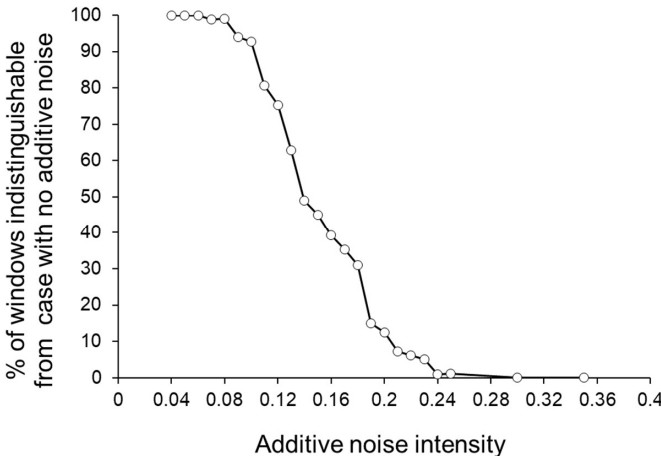





Fig. 5. Percentage of 50 data windows, shifted by 50 data step, of crack fusion model with added noise
indistinguishable from the initial condition (with no added noise).

Once we been convinced that our data analysis is reliable for the targeted research goal, we
continued analysis of catalogue data. First of all, we calculated MD values for 50 data windows
shifted by 1 data (Fig. 6).

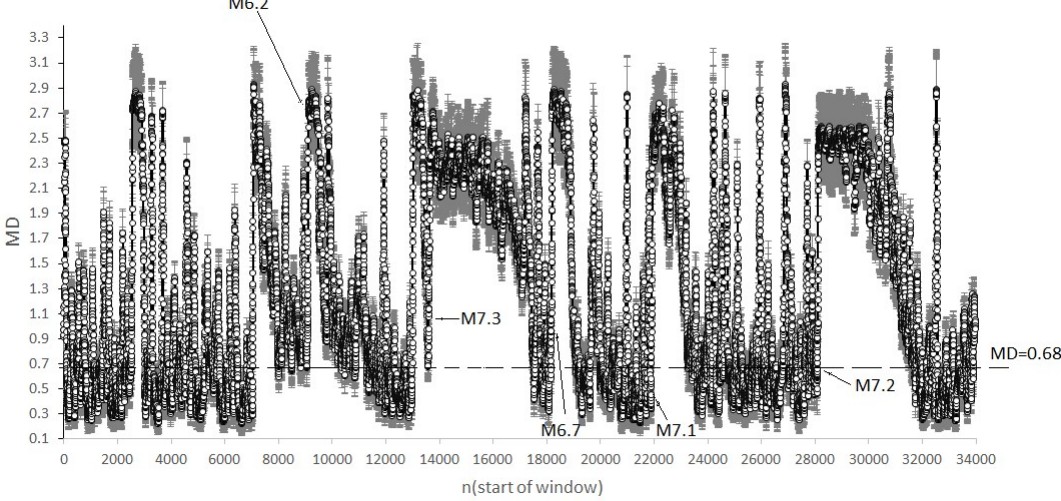



Fig. 6. Averaged MD values calculated by comparing *ICT(i), ICD(i)* and *ICE(i)* sequences from the original
SC catalogue and from the set of randomized catalogues. Dotted line corresponds to significant difference
between windows at p=0.05. MD values are calculated for 50 data windows shifted by 1 data.

Results in figures 2 and 6, are in agreement with the view that in spite of generality of
background physics [Lombardi&Marzocchi, 2007; Di Toro et al, 2004; Davidsen&Goltz, 2004;
Helmstetter&Sornette, 2003; Corral, 2008] we observe two separate processes prior and after main
shocks [Sornette&Knopoff, 1997; Davidsen&Goltz, 2004; Wang&Kuo, 1998]. According to
recent views, latest one is characterized by the long and short range correlations and thus is more
ordered, while the former apparently is more uncorrelated or random-like [Touati et al. 2009;
Godano, 2015]. Indeed, according to Bowman et al. [2004] loss of energy (released also in the
form of seismic energy) related with the occurrence of strong event, introduces memory into the
system [Bowman&Sammis, 2004]. We see in Figs. 2 and 6, that seismic process assessed by
*ICT(i), ICD(i)* and *ICE(i)* variability after strongest regional earthquakes is clearly different from
randomized catalogues and thus is more regular comparing to periods prior strong events. In
addition to this, it is noticeable that in 33% of all considered 50 data windows (usually prior to
strongest earthquakes), original seismic process is indistinguishable from randomised catalogues.
In order to exclude that some characteristic, out of selected three ones (*ICT(i), ICD(i)* and
*ICE(i)*), influence obtained results more than others, we accomplished similar analysis comparing
groups of original and randomized catalogues by two characteristics. Results of such analysis (not
shown here) of separate comparison of groups consisted by pairs of *ICT(i)* and *ICD(i), ICT(i)* and



*ICE(i), ICD(i)* and *ICE(i)* columns, generally coincide with the results of above analysis
(accomplished for groups consisted by all three columns). This convinces that results of our
analysis can not be reduced to the influence of only one single characteristics. Thus, changes in
Figs. 2 and 6, reveal changes in dynamical features of seismic process as whole, involving changes
in all three its domains.
Next, for better visibility of above results (see Fig. 6), in Fig. 7, we present MD values
calculated for 50 data windows in period from 14.05.1990 (window started from event 12100 in
considered SC catalogue) to 28.06.1992 (window started from event 13797 in SC catalogue). In
this period two strongest earthquakes M6.1 (23.04.1992) and  M7.3 (28.06.1992) occurred. Prior
to both strong earthquakes we observe windows in which seismic process by variation of *ICT(i),*
*ICD(i)* and *ICE(i)* data is indistinguishable from randomized catalogues (see circles below dotted
significant difference line). Also, it is noticeable that after these strong events, extent of order in
seismic process, according to changes in MD values, strongly increases (original catalogue
becomes stronger different from randomized catalogue). In case of M7.3 such increase lasted for
considerably long time after strong event, at least till about January of 1993 (see Fig. 6).

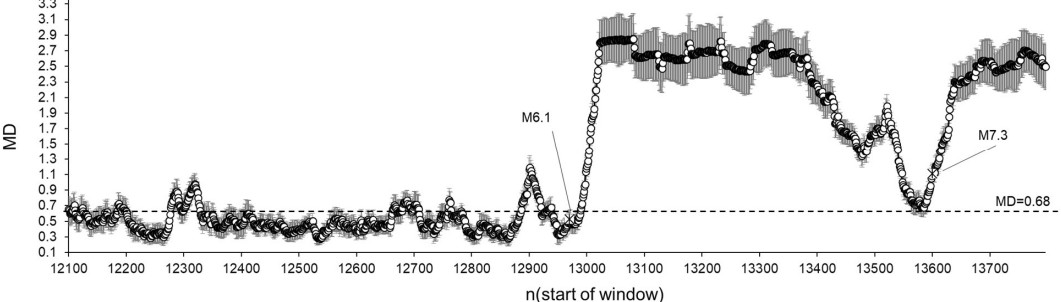


Fig. 7. Averaged MD values calculated for period from 14.05.1990 (12100) to 28.06.1992 (13797) where
two strongest earthquakes occurred M6.1 (23.04.1992) and M7.3 (28.06.1992). MD are calculated by
comparing *ICT(i), ICD(i)* and *ICE(i)* sequences from the original SC catalogue and from the set of
randomized catalogues. Dotted line corresponds to significant difference between windows at p=0.05. MD
values are calculated for 50 data windows shifted by 1 data.
Next period which we selected for detailed analysis elapsed from 24.08.97 (window started
from event 20760 in the used SC catalogue) to 16.10.99 (window started from the event 21160 in
SC catalogue). Strongest earthquakes occurred in this period are M5.23 (06.03.1998) and M7.1
(16.10.1999). Results presented in Fig .8, are mostly similar to what we see in Fig. 7. Exactly,
strongest earthquakes are preceded by windows in which seismic process in original catalogue is
indistinguishable from randomized catalogues. After strong earthquakes, seismic process in the
original catalogue, according to features of simultaneous variations of *ICT(i), ICD(i)* and *ICE(i)*
characteristics, is strongly different from random process.

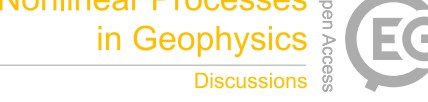

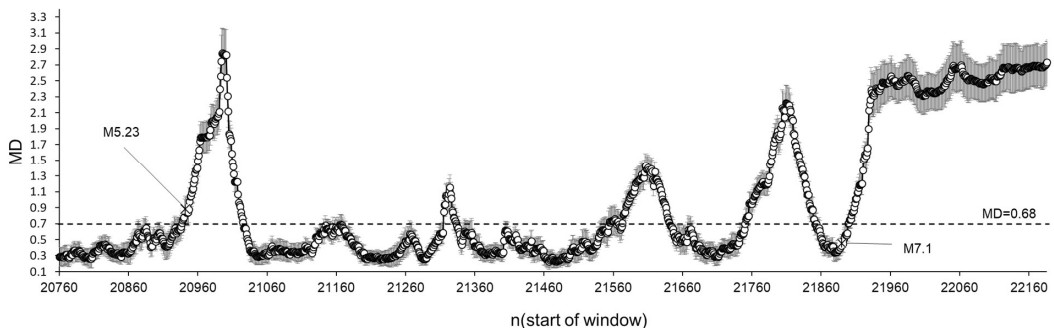


Fig. 8. Averaged MD values calculated for period from 24.08.97 (20760) to 16.10.99 (21160) where two
strongest occurred earthquakes are M5.23 (06.03.1998) and M7.1 (16.10.1999). MD are calculated by
comparing *ICT(i), ICD(i)* and *ICE(i)* sequences from the original SC catalogue and from the set of
randomized catalogues. Dotted line corresponds to significant difference between windows at p=0.05. MD
values are calculated for 50 data windows shifted by 1 data

     Separate consideration of situation for period including strongest M7.2 earthquake leads to
similar conclusion. In Fig. 9, we again observe that prior to strong earthquakes, seismic process
looks mostly like random and that extent of order strongly increase after these events.
     As it was expectable, in this sense, behavior of seismic process prior and after all
considered strong events is similar, only difference is the length of the period during which post-
earthquake seismic process remain significantly regular comparing to randomized catalogues. For
strongest earthquakes this period is clearly longer (see Fig. 6). This was quite logical and
apparently is connected with the generation of series of aftershocks which spatial, temporal and
energetic features are causally related with the mainshock. This is in agreement with well known
productivity law, stating that the larger the mainshock magnitude the larger is the total number of
aftershocks [Helmstetter, 2003; Godano, C., Tramelli, 2016]. Here need to be underlined that the
question of the temporal length of aftershock sequence following strong earthquake, is still not
understood because is related with the problem of time scale of background seismic activity,
becoming again dominant with respect to the rate of aftershocks' occurrence [Godano, C., Tramelli,
2016].





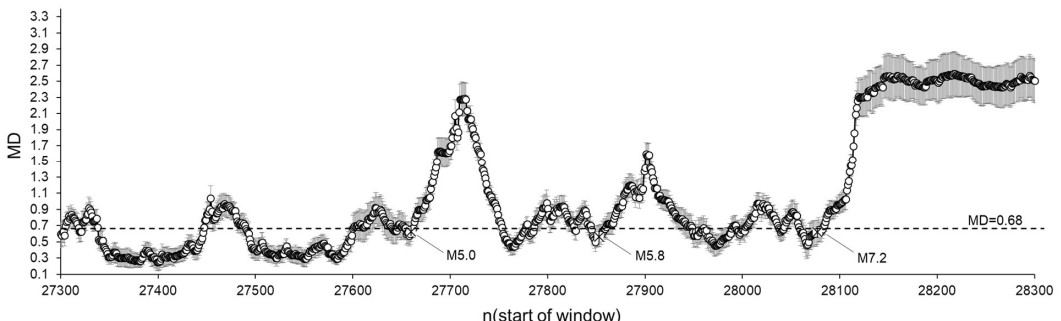

Fig. 9. Averaged MD values calculated for period from 30.10.2008 (27300) to 05.04.2010(28300) where three strongest occurred earthquakes are M5.0 (01.10.2009), M5.8(30.12.2009) and M7.2(04.04.2010). MD are calculated by comparing *ICT(i), ICD(i)* and *ICE(i)* sequences from the original SC catalogue and from the set of randomized catalogues. Dotted line corresponds to significant difference between windows at p=0.05. MD values are calculated for 50 data windows shifted by 1 data.

From results in Figs. 7-9, it can be said that the extent of the order in seismic process (assessed by features of earthquakes temporal, spatial and energetic distribution) may be changed not only in the periods prior and after of strongest (M7.3, M7.2 and M7.1) earthquakes, but also prior and after other (not strongest) events too. Example, as we see in windows from 21570 to 21770 (Fig. 8), pairs of earthquakes occurred in about two weak periods (M4.93, 14.05.1999 and M4.92, 01.06.1999 as well as M4.71, 24.08.1999 and M4.8, 10.09.1999) also cause increase in the extent of order of seismic process. Similar is conclusion from Figs. 7 and 9. Most important in all cases still is the fact that the increase in the extent of order occurs after strong earthquakes, while prior to these events, in periods which can be regarded as relatively calm, original seismic process remains not distinguishable from the random process, assessing it by the variation of *ICT(i), ICD(i)* and *ICE(i)* data.

Since, based on above results, we suggested that prior to strong earthquakes seismic process of relatively small (with M<4.6, [Hough, 1997]) earthquakes' generation is random-like, it was necessary to analyze additionally the behavior of small earthquakes which occur after strong events. For this we selected periods of relatively small seismic activity involving events with magnitudes M ≤ 4.6. Exactly, 2-5 days periods of less than M4.6 aftershock activity, soon after strong earthquakes, have been considered. Results of analysis for three such periods followed strongest M7.3, M7.1 and M7.2 earthquakes are presented in Figs. 10-12. As follows from these figures we do not observe windows in which original seismic process, according to distribution of its *ICT(i), ICD(i)* and *ICE(i)* characteristics, can be regarded as similar to randomized catalogues. In all three analyzed cases in period of clear aftershock activity, immediately after strong earthquakes, in all windows seismic process is strongly different from the set of randomized catalogues. In other words, in the original catalogue, seismic process after strong events in periods of relatively small (M ≤ 4.6) earthquakes generation is significantly regular comparing to randomized catalogues. It can be added here that the similar situation was for sequences of small earthquakes occurred after strong, but not strongest, earthquakes (e.g. M.6.0). All this also convinces that in the periods of aftershock activity original seismic process is strongly different from what is observed for randomized catalogues in which we distorted spatial, temporal or energetic distribution features.



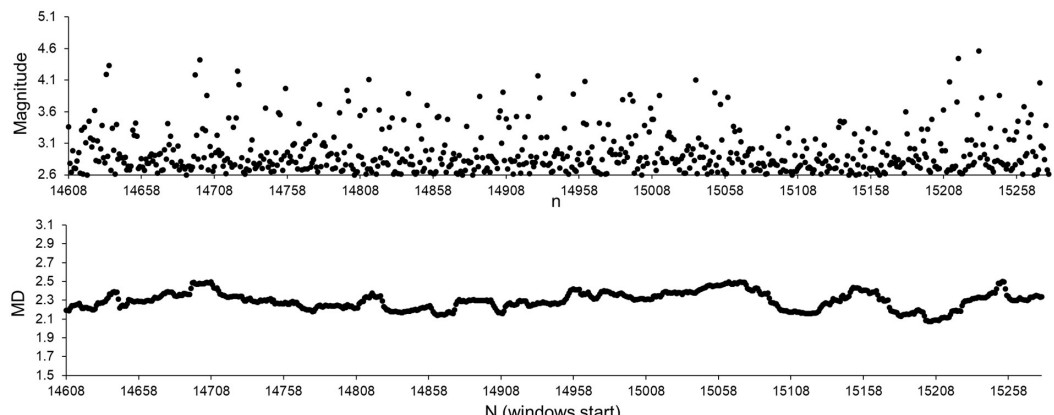

Fig. 10. Magnitudes and MD values calculated for part of SC catalogue after M7.3 (28.06.1992, sequential number in SC catalogue 13648) from 01.07.1992 (sequential number in SC catalogue 14608) to 05.07.92 (sequential number in SC catalogue 15280). Average MD values are calculated for 50 data windows, shifted by 1 data.

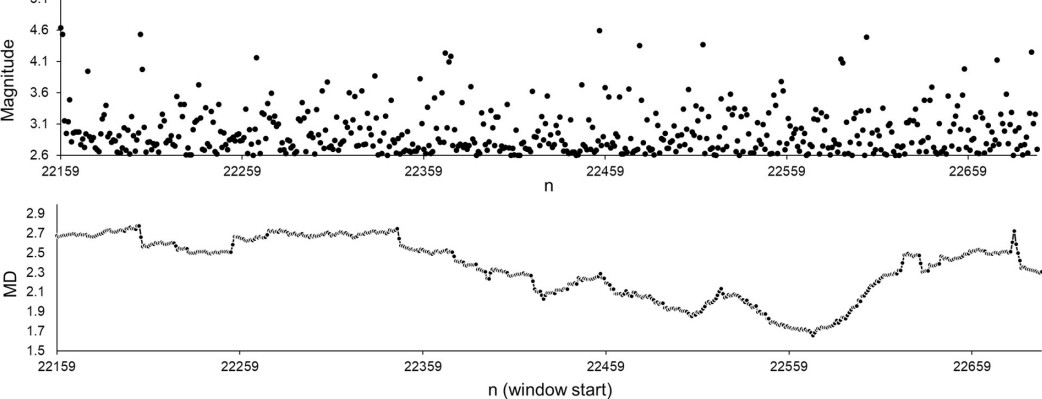

Fig. 11. Magnitudes and MD values calculated for part of SC catalogue after M7.1 (16.10.1999, sequential number in SC catalogue 21937) from 16.10.1999 (sequential number in SC catalogue 22159) to 21.10.1999 (sequential number in SC catalogue 22697). Average MD values are calculated for 50 data windows, shifted by 1 data.



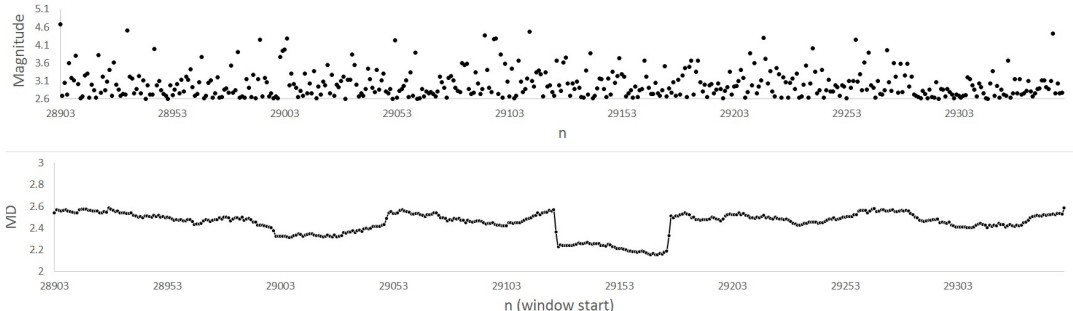

Fig. 12. Magnitudes and MD values calculated for part of SC catalogue after M7.2 (04.04.2010, sequential number in SC catalogue 28129) from 06.04.2010 (sequential number in SC catalogue 28903) to 08.04.2010 (sequential number in SC catalogue 29350). Average MD values are calculated for 50 data windows, shifted by 1 data.

Next we accomplished similar analysis for the sequences of relatively small earthquakes occurred in periods when no strong earthquakes have been registered. These small earthquakes apparently can not be regarded as aftershocks of strong events. Indeed, in Fig. 13, analyzed almost two year period of small earthquakes activity has started 5 month later after M5.12 earthquake which was closet earthquake exceeding the selected M4.6 threshold. According to present views about aftershocks time distribution it looks very unlikely that M5.12 earthquake could invoke aftershock activity which lasted two years. Thus, in agreement of our above findings we can conclude that, for selected period, in 60% of considered 50 data windows the seismic process, in the original catalogue, looks indistinguishable from the randomized by shuffling procedure set of catalogues.

In Fig. 14, we present results for the next part of catalogue containing relatively small earthquakes in the observation period which is far from occurrence times of strongest events. Relatively strong earthquake M5.43 (07.07.2010, sequential number in SC catalogue 31011) occurred 9 month prior to the start of this 10 month long period of small earthquake activity which lasted from 07.04.2011(sequential number in SC catalogue 31823) to 14.02.2012 (sequential number in SC catalogue 32240). In this case we observe that in 75% of analyzed 50 data windows, the seismic process in original catalogue is indistinguishable from the set of randomized catalogues.



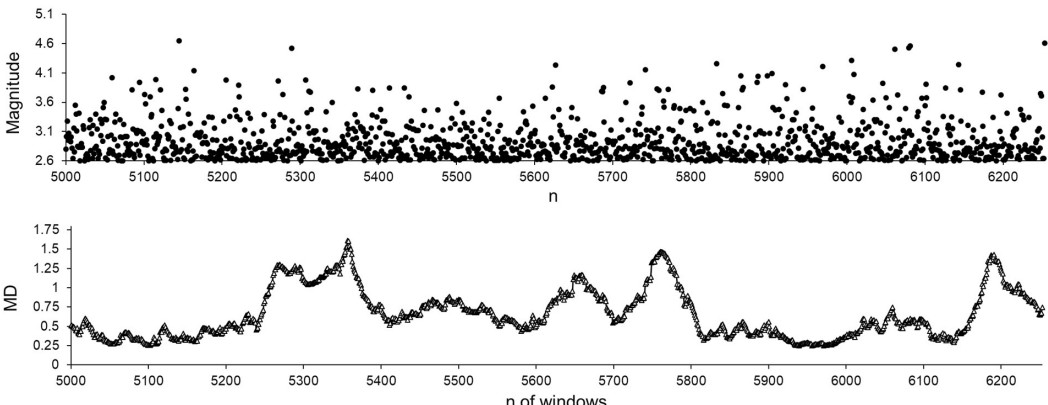

Fig. 13. Magnitudes and MD values calculated for non aftershock part of SC catalogue from 07.03.1983 (sequential number in SC catalogue 5000) to 05.02.1985 (sequential number in SC catalogue 6253). Average MD values are calculated for 50 data windows, shifted by 1 data.

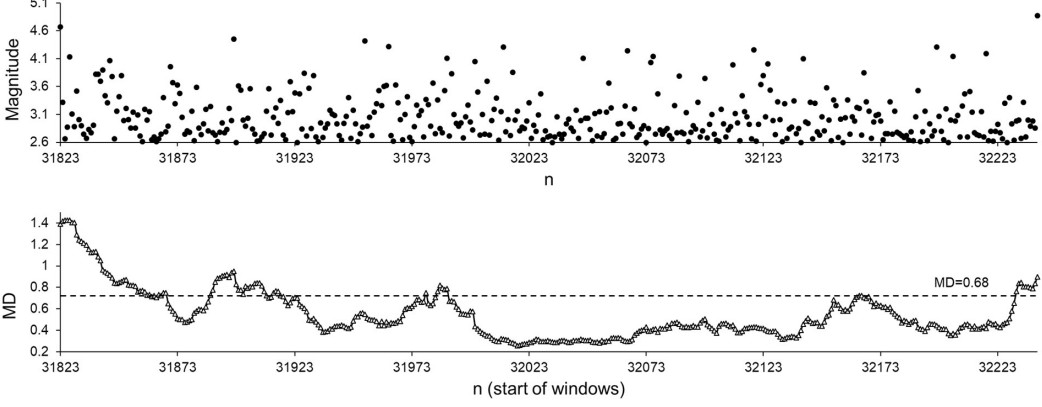

Fig. 14. Magnitudes and MD values calculated for non aftershock part of SC catalogue from 07.04.2011 (sequential number in SC catalogue 31823) to 14.02.2012 (sequential number in SC catalogue 32240). Average MD values are calculated for 50 data windows, shifted by 1 data.

In Fig. 15, we present results for the third part of catalogue which also was selected so that contained relatively small earthquakes, M≤ 4.6, in period far from strongest events (closest such earthquake M7.1 occurred more than 5 year earlier, on 16.10.1999, sequential number in considered SC catalogue is 21937). Two relatively strong M5.7 earthquakes (08.12.2001 and 22.02.2002 with sequential numbers in SC catalogue 24491 and 24640) also occurred essentially long before selected period which lasted from 24.05. 2006 to 05.08.2007. In this period of



generation of small earthquakes, 84% of 50 data windows indicated that seismic activity in original catalogue is indistinguishable from the set of randomized catalogues.

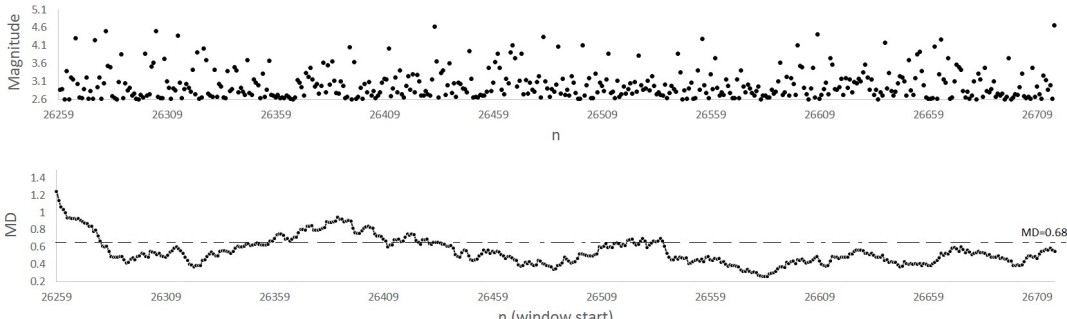

Fig. 15. Magnitudes and MD values calculated for non-aftershock part of SC catalogue from 24.05. 2006 (sequential number in SC catalogue 26259) to 05.08.2007 (sequential number in SC catalogue 26717). Average MD values are calculated for 50 data windows, shifted by 1 data.

As we have seen from above results, assessed by $ICT(i)$, $ICD(i)$ and $ICE(i)$ characteristics, seismic process of relatively small earthquakes generation not always looks random-like and strongly depends on the space and time location of such small earthquake sequences. It can be supposed that if observed indistinguishability from the randomness really is connected with features of seismic process in periods preceding strongest events, then such indistinguishability should be retained for higher representative threshold values too. To test this assumption, we accomplished the same analysis for southern California earthquake catalogues with the representative thresholds M3.6 and M4.6. Further increase of threshold had no sense because only 29 of such earthquakes occurred for considered period.

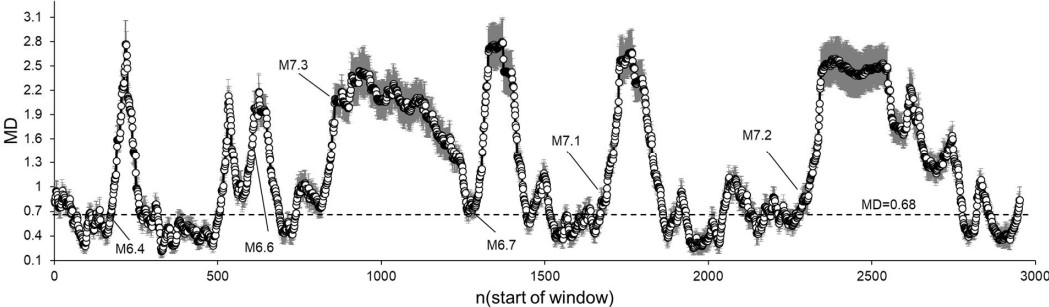

Fig. 16. Averaged MD values calculated by comparing $ICT(i)$, $ICD(i)$ and $ICE(i)$ sequences from the original SC catalogue and from the set of randomized catalogues(representative threshold M3.6). Dotted line corresponds to significant difference between windows at p=0.05. MD values are calculated for 50 data windows shifted by 1 data.



In Fig. 16, we give results for representative threshold M3.6. We see that situation with
windows in which seismicity is indistinguishable from randomness is almost completely similar
to what is presented in Fig. 6, for representative threshold M2.6. Exactly, in 33% of all 50 data
windows seismic process looks similar with random process in catalogues where dynamical
structure of original seismic process was intentionally distorted. These random-like windows in
original catalogue preceded strongest occurred in the same catalogue events.

Most interesting was analysis at further increase of representative threshold (to M4.6)
below which, as it was said above, we regarded earthquakes as small [Hough, 1997].

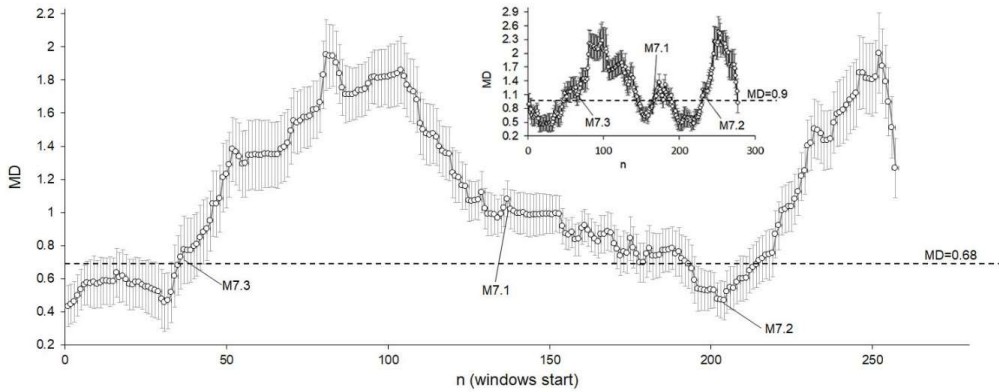


Fig. 17. Averaged MD values calculated by comparing *ICT(i), ICD(i)* and *ICE(i)* sequences from the
original SC catalogue and from the set of randomized catalogues(representative threshold M4.6). Dotted
line corresponds to significant difference between windows at p=0.05. MD values are calculated for 50 data
windows shifted by 1 data. In the inset are presented results calculated for 30 data windows shifted by 1
data step.

As we see in Fig. 17, in case of high representative threshold M4.6, prior to two strong
events, M7.3 and M7.2, we observe windows (of 50 data) in which seismic process, by the
variability of *ICT(i), ICD(i)* and *ICE(i)* characteristics, is indistinguishable from randomized
catalogues. In total 21% of all, 50 data, windows indicated calculated MDs lower than significance
threshold value (0.68). On the other hand, at high representative threshold (M4.6), in different
from above cases, prior to strong M7.1 earthquake, we do not observe 50 data windows in which
seismic process could be regarded as random.

This apparently is caused by the small amount of events above M4.6 threshold in catalogue
and by the selected length of window (50 data) for mentioned small data sequence. Indeed, in the
case of 30 data windows shifted by 1 data, we see that prior to M7.1 there also are windows
indistinguishable from random catalogues (see inset in Fig. 17). Percentage of such windows with
random behavior of seismic process is 37. Commenting results in Fig.17, we can say that shorter
windows (apparently in the range 30-50 data) look preferable for analysis like carried out in this
work, and that randomlike character of seismic process in windows prior to strong events, is not
connected  only with small earthquakes.

Based on all above analysis we conclude that seismic process in general, can not be
regarded neither as completely random or as deterministic. The dynamics of the seismic process

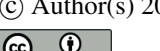


undergoes strong time depending changes. In other words, the extent of regularity of seismic process, assessed by features of temporal, spatial and energetic distributions, is changing over time what is in complete accordance with time-dependent variations proposed by intermittent criticality concept of earthquake generation.

In some periods seismic process looks closer to randomness while in other cases it becomes closer to regular behavior. Exactly, in periods of relatively decreased earthquake generation activity (at smaller energy release), seismic process looks random-like while in periods of occurrence of strong events, followed by series of aftershocks, it reveal significant deviation from randomness - the extent of regularity essentially increases. The period, for which such deviation from the random behavior can last, depends on the amount of seismic energy released by the strong earthquake. Found results on multivariable assessment of dynamical features of seismic process are in accordance with our previous findings on dynamical changes of earthquakes temporal distribution [Matcharashvili et al. 2018].

## Conclusions

We have investigated variability of regularity of seismic process based on its spatial temporal and energetic characteristics. For this purpose we used southern Californian earthquake catalogue from 1975 to 2017. The method of analysis represented combination of multivariate Mahalanobis distance calculation with the surrogate data testing. We accomplished the multivariate assessment of changes in the extent of the regularity of seismic process, based on increments of cumulative times, increments of cumulative distances and increments of cumulative seismic energies, calculated from southern California earthquake catalogue.

In order to assess the ability of the used multivariate approach to discriminate different conditions of dynamical systems we used 3 dimensional models in which dynamical features were changed from more regular to the more randomized conditions by adding some extent of noises.

It was shown that in about third part of considered 50 data windows, the original seismic process is indistinguishable from random process by its features of temporal, spatial and energetic variability. Prior to strong earthquake occurrences, in periods of relatively small (<M4.6) earthquakes generation, percentage of windows in which seismic process is indistinguishable from random process essentially increases (to 60-80%). At the same time, in periods of aftershock activity in all considered windows the process of small earthquake generation become regular and thus is strongly different from randomized catalogues.

According to results of analysis we conclude that seismic process in general, can not be regarded neither as completely random or as completely regular (deterministic). Instead, we can say that the dynamics of the seismic process undergoes strong time depending changes. In other words, the extent of regularity of seismic process, assessed by features of temporal, spatial and energetic distributions, is changing over time.

Also it was shown that in some periods seismic process looks closer to randomness while in other cases it becomes closer to regular behavior. Exactly, in periods of relatively decreased earthquake generation activity (at smaller energy release), seismic process looks random-like while in periods of occurrence of strong events, followed by series of aftershocks, it reveal significant deviation from randomness - the extent of regularity essentially increases. The period, for which such deviation from the random behavior can last, depends on the amount of seismic energy released by the strong earthquake.

## Acknowledgements



This work was supported by Shota Rustaveli National Science Foundation (SRNSF), grant 217838
"Investigation of dynamics of earthquake's temporal distribution".

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
