# Peer review of "Mahalanobis distance-based recognition of changes in the dynamics of a seismic process"

_Nonlinear Processes in Geophysics, 2018_

## Referee Comment (RC1) · Eleftheria Papadimitriou (Referee) · 15 Mar 2019

The paper deals with the recognition of changes in the dynamics of the earthquake process, as it is clearly stated by the title per se, by estimating the Mahalanobis distance (MD) in Southern California, an earthquake prone area, and for this reason is important. The importance is connected with the identification of regularities in the seismicity behavior during periods of seismic excitations, whereas it turns to random – like during periods of decreased activity. There are, however, several points inside the paper that need additional work and corrections. Specific comments are reported, which I hope will contribute to the improvement of its revised version.

MAJOR COMMENTS 1. Considerable improvement of syntax and grammar is re-

quired. In some places the reader needs to "infer" the meaning of the content. 2. Lines 91 and 92: This point is crucial for the paper and even more for the reader. It is necessary, therefore, to be more specific and more explanation to be given, how this goal will be approached and what is expected from the analysis. 3. It is important to see the details of the plots commented and determined. For example, in Fig. 3, the MD variation and the connection with the strong earthquakes, shown in the upper part of the figure. 4. Lines 301 – 302: ". . . we observed two separate processes prior and after main shocks . . .". Is this observed only in the case of the strong earthquakes' occurrence or not? And how it could be determined in the cases that the same pattern is met without a strong earthquake occurrence? 5. Lines 322 – 323: Why did you chose this period and not repeat the exercise for all M6.0 earthquakes shown in Figure 3? 6. In Figure 7, the increase around 12300 is also profound, as before the M6.1 event. How do you determine that? 7. Line 340: earthquake of M5.23, a moderate magnitude event. Could you chose and suggest a threshold for the strong events, which for the pattern is seeking? Could you present then the catalog of all these events and for which the pattern is observed and is statistically significant? 8. Line 345: ". . . is strongly different from random process . . ." Could you clarify this statement in relation with details in Figs 7 & 8? 9. Fig. 8: How is the pattern before and after the M5.23 earthquake and how is it explained? 10. Fig. 8: What happened before and after the M7.1 earthquake, how is it determined and how is it compared with the corresponding behavior connected with the M7.3 of Fig. 7? 11. Lines 362 – 365: Please, clarify your statement and provide arguments to support it. 12. Line 392: ". . . 2 – 5 days period . . ." How this period has been set? This need to be supported and clearly stated how is it associated with the aftershock activity evolution.

SPECIFIC COMMENTS 1. Special caution should be paid to the citation, since a unique format is not followed. For example, in page 2 the same paper is written as: "Iliopoulos et al., 2012" and "Iliopoulos, et al. 2012". For this citation in particular, you need to correct in line 610: Instead of "Papadimitriou, P. P." the correct name is "Papadimitriou, E. E.". More: when the authors names are shown inside the text

the commas should be avoided, like in lines 135 & 136: Kanamori [1977] (without a comma). Line 306: Bowman and Sammis (written properly some lines below), instead of Bowman et al. Lines 362 & 366: Please, be kind enough to correct the citation format. 2. Page 3: The map should be limited to the boundaries of the catalog – it seems now that seismicity has ceased (there is no seismicity) northern than 38 degrees for example, or to the west of -122 degrees. 3. It could be of broad interest and concern of many readers to see why the authors did not prefer to use seismic moment, which is nowadays routinely estimated, instead of seismic energy. 4. Syntaxis in many places needs substantial review, for the text to be conceivable. 5. When you refer to "strong" earthquake, please, pay attention to not name them "strongest" (it is met in many places in the manuscript) 6. Lines 402 – 404: Could you make it more clear? 7. Line 476: "... 29 of such earthquakes occurred for considered period ...": What do you mean by that?

---

## Author Comment (AC1) · 29 Mar 2019

*The paper deals with the recognition of changes in the dynamics of the earthquake process, as it is clearly stated by the title per se, by estimating the Mahalanobis distance (MD) in Southern California, an earthquake prone area, and for this reason is important. The importance is connected with the identification of regularities in the seismicity behavior during periods of seismic excitations, whereas it turns to random – like during periods of decreased activity. There are, however, several points inside the paper that need additional work and corrections. Specific comments are reported, which I hope will contribute to the improvement of its revised version.*

MAJOR COMMENTS

*1. Considerable improvement of syntax and grammar is re-quired. In some places the reader needs to "infer" the meaning of the content.*

After the discussion stage the improved manuscript will be checked and corrected by a language expert.

*2.Lines 91 and 92: This point is crucial for the paper and even more for the reader. It is necessary, therefore, to be more specific and more explanation to be given, how this goal will be approached and what is expected from the analysis.*

To make more understandable what was the goal and what we expected, Lines 91 -92 and further were rewritten as follows:
Such multivariate comparison of original catalogue with randomized catalogues may help to gain new knowledge about the character of changes which definitely occur in the extent of order/disorder of seismic process. Besides, we will have stronger arguments to speak where and how the dynamics of original seismic process in analyzed catalogue was close to disorder (irregularity) or to order (regularity). We also may find out how such changes may be related with the process of strong earthquakes preparation.

Obtained in our research results show that the extent of regularity in the analyzed seismic process is changing, being closer to randomness in periods prior to strong earthquakes. After strong earthquakes, the regularity of original seismic process assessed by used temporal spatial and energetic characteristics is clearly increased.

*3. It is important to see the details of the plots commented and determined. For example, in Fig. 3, the MD variation and the connection with the strong earthquakes, shown in the upper part of the figure.*

In the upper part of Fig.3 we present calculated for consecutive windows seismic energies and averaged MD values in the bottom. Time scale or the numeration of windows for upper and bottom plots is the same. So, in Fig.3 we wanted to make easily comparable the time (window) location of strong earthquakes and the location of time periods (windows) when seismic process according to our results was closer to regular or random behavior.
Dotted line in figures shows MD value which corresponds to critical Fc=3.99 (according to statistical tables for given degrees of freedom, Fc=3.99 corresponds to significant difference between groups at p=0.01). Thus if MD value for given window is larger than MD=0.68 (i.e. if F value for this window >3.99) then this window is significantly (at p=0.01) different from randomized windows. We underline here that by mistake in our manuscript p=0.05 was shown what is corrected in revised version.

*4. Lines 301 – 302: ": : : we observed two separate processes prior and after main shocks : : :". Is this observed only in the case of the strong earthquakes' occurrence or not? And how it could be determined in the cases that the same pattern is met without a strong earthquake occurrence?*

We are sorry for the unclear text. Here we have just meant the fact (known from literature) about differences between processes prior and after strong events, based on authors cited in followed text. In the revised version the text related to this remark is in our opinion more clear. In addition to the text in manuscript we add here that we do not state that windows with smaller earthquakes will always be random-like; what we would like to say is that portion of such windows after strong earthquakes is zero, while prior to strong earthquakes increase at least to 33% of all windows.

*5. Lines 322 – 323: Why did you chose this period and not repeat the exercise for all M6.0 earthquakes shown in Figure 3?*

As it is pointed in the manuscript, we show in Figs. 6, 7, 8, certain periods selected from the Fig. 3 just for better visibility of observed changes. Thus, we have chosen a part of the catalogue with the strongest observed earthquake and the preceding smaller ones. As for other M > 6.0 earthquakes we think that changes are well visible in Fig. 3, for the case of 50 data windows shifted by 50 data and we did not want to overburden the text with figures with similar results.

*6. In Figure 7, the increase around 12300 is also profound, as before the M6.1 event.*
*How do you determine that?*

What can be said for sure is that after strong EQs we observe a largely increase number of windows in which seismic process looks as regular. Number of the windows is larger for stronger EQs. This apparently is related with the correlated aftershock activity. For smaller EQs such period is smaller. The fact is that number of windows with random like behavior of seismic process essentially increases for periods of smaller EQ activity.

*7. Line 340: earthquake of M5.23, a moderate magnitude*
*event. Could you choose and suggest a threshold for the strong events, which for*
*the pattern is seeking? Could you present then the catalog of all these events and*
*for which the pattern is observed and is statistically significant?*

After reviewer's remark we changed and expanded text after Fig.7 in following manner: There are two large events occurred in this period: the moderate M5.23 (06.03.1998) and the strong M7.1 (16.10.1999) earthquakes. Here we underline the obvious fact that there is no use to try to find a magnitude range which may occur in windows where seismicity behaves random-like. Indeed, as we see from our results (see Fig. 3, 6 and 7), earthquakes of any size may occur in any windows both in those where seismic process is closer to regular behavior or where it is more random-like. So, we cannot speak about the magnitude threshold or about the range of magnitudes in the sense of their immediate influence on changes in the extent of regularity of seismic process. On the other hand, our results show that in periods of mostly small earthquakes generation, prior to strong earthquake occurrence, seismic process in the considerable amount of windows is indistinguishable from randomness. Thus, as assessed by simultaneous variations of *ICT(i), ICD(i)* and *ICE(i)* characteristics the seismic process of relatively small earthquakes generation prior to strong earthquakes can be regarded as random-like.

*8. Line 345: ": : : is*
*strongly different from random process : : :" Could you clarify this statement in relation*
*with details in Figs 7 & 8?*

We base our statement on the results of our analysis showing that MD is larger than the threshold value what means that seismic process in these windows should be regarded as different from random process. Corrected text after reviewer's remark reads as following:
Exactly, strong and relatively strong (for selected short period) earthquakes are preceded by considerable amount of windows in which seismic process, in the original catalogue, is indistinguishable from what we observe for randomized catalogues. Contrary to this, in all (50 data) windows followed strong (or relatively strong) earthquakes, we observe statistically significant difference of original seismic process from processes taking place in randomized

catalogues. Indeed, the multivariate comparison of these windows, based on variation of ICT(i), ICD(i) and ICE(i) characteristics, convinces us that in these windows seismic process is strongly different from random process (see Figs. 7 and 8).

*9. Fig. 8: How is the pattern before and after the M5.23 earthquake and how is it explained?*

As we mentioned above we had not try to find magnitude range of events which may occur in windows where seismicity behaves random-like or regular. Apparently at present we cannot answer this question. Based on our results we see that, some strong EQs occur in random-like windows while others occur in regular windows. Also, small EQs occur both in more regular as well as in the more random-like windows. Apparently all this is related with presently unknown background physics of earthquake generation and related changes in the long range features of seismic process. In present work we aimed to provide additional arguments in favor of based on recent researches view that the degree of regularity (randomness) of seismic process undergoes essential changes which can be detected and even assessed quantitatively.

*10. Fig. 8: What happened before and after the M7.1 earthquake, how is it determined and how is it compared with the corresponding behavior connected with the M7.3 of Fig. 7?*

Most 50 data windows prior to M7.1 indicate a random-like behavior (MD<threshold value) but after, in a number of windows we observe the pattern significantly different from a random behavior. In this sense results in Fig.7 and 8 are in fact similar. It can be supposed that increase in MD values observed prior M7.1 and M7.3 may be related with foreshock activity. At the same time we point that in the present research we want just to present new results on a changing extent of regularity, what someway should be related with the foreshock activity during strong earthquakes preparation.

*11. Lines 362 – 365: Please, clarify your*
*statement and provide arguments to support it.*

We are grateful for this remark. Here we have cited works of other colleagues. Statement in lines 362-365 is that: "….. aftershocks spatial, temporal and energetic features are causally related with the mainshock". This well known and accepted view is based on facts which often were described in related literature. In the manuscript we provide some, among many others, references, which agree with statement in our manuscript. In our opinion presented results agree with this vision.

*12. Line 392: ": : : 2 – 5 days period*
*: : :" How this period has been set? This need to be supported and clearly stated how*
*is it associated with the aftershock activity evolution.*

We have selected (mentioned in manuscript) 2-5 days period followed soon after strongest M7.3, M7.1, and M7.2 earthquakes. We based our selection on the fact that the larger is main shock magnitude the larger is the number of aftershocks. Also the selected period started soon (1 to 3 days) after three strongest catalogue earthquakes when aftershock activity could not be ended. Moreover, we calculated distances from these aftershocks to corresponding main shock and find that they are localized in close vicinity to main shocks. Exactly, 90% of earthquakes in this 2-5 days period occurred in 0.5-70km distance from epicenter of M7.3; 92% of earthquakes in this period occurred in 1.2-60km distance from epicenter of M7.1; . 99% of earthquakes in this period occurred in 0.7-60km distance from epicenter of M7.2. All this has convinced us that we mostly deal with small earthquakes related to the mainshocks aftershock activity. We could also show maps of these earthquakes location but finally we abandoned this idea in order to not overburden manuscript with not too informative figures.

SPECIFIC COMMENTS

*1. Special caution should be paid to the citation, since a*
*unique format is not followed. For example, in page 2 the same paper is written as:*
*"Iliopoulos et al., 2012" and "Iliopoulos, et al. 2012". For this citation in particular,*
*you need to correct in line 610: Instead of "Papadimitriou, P. P." the correct name*
*is "Papadimitriou, E. E.". More: when the authors names are shown inside the text the commas*
*should be avoided, like in lines 135 & 136: Kanamori [1977] (without a*
*comma). Line 306: Bowman and Sammis (written properly some lines below), instead*
*of Bowman et al. Lines 362 & 366: Please, be kind enough to correct the citation*
*format.*

We are grateful to reviewer for this remark and have tried to correct all inaccuracies with citations.

*2. Page 3: The map should be limited to the boundaries of the catalog – it*
*seems now that seismicity has ceased (there is no seismicity) northern than 38 degrees*
*for example, or to the west of -122 degrees.*

New map in revised version is better limited by boundaries of the catalog.

*3. It could be of broad interest and concern of many readers to see why the authors did not prefer*
*to use seismic moment, which is nowadays routinely estimated, instead of seismic energy.*

It is pointed in the manuscript that we aim at assessment of dynamical changes in entire seismic process based on all its domains, temporal, spatial and energetic. Thus, it was logical to use seismic energy in our research. On the other hand, we had not special reasons to go in discussion regarding the using of seismic energy and seismic moment, characteristics which both are calculated from earthquakes magnitudes, in researches of other authors. Moreover in this work we preferred to use energy values calculated from magnitudes because we usually did so in our last researches and

it looked logical to use in this research the same characteristic. This of course do not mean that we question using of seismic moment by other authors and in future works we also will use seismic moments in cases when it will look appropriate for certain research goal.

*4. Syntaxis in many*
*places needs substantial review, for the text to be conceivable.*

Thanks for this comment, we have done our best to correct the text.

*5. When you refer to*
*"strong" earthquake, please, pay attention to not name them "strongest" (it is met in*
*many places in the manuscript).*

Many thanks for useful remark. In revised version text is corrected.

*6. Lines 402 – 404: Could you make it more clear?*

Here we just mean that during the period of aftershock activity seismic process looks more regular and significantly different from random process comparing to the period prior earthquake; apparently this is happening not only in case strongest but also smaller and even moderate earthquakes too.
This part of manuscript after Fig. 9, is now considerably rewritten.

*7. Line 476: ": : : 29 of such earthquakes occurred for considered period : : :": What do*
*you mean by that?*

Thanks for remark. Now the text reads: Further increase of threshold to M5.6 had no sense because only 29 of such earthquakes, i.e. with magnitudes larger than M5.6, occurred in south Californian catalogue for the period considered in this research.

---

## Referee Comment (RC2) · Antonella Peresan (Referee) · 28 Apr 2019

**Comments on**
***"Mahalanobis distance based recognition of changes***
***in the dynamics of seismic process"***
by Teimuraz Matcharashvili, Zbigniew Czechowski, Natalia Zhukova

A novel method is proposed, based on well-established Mahalanobis metric, to quantitatively assess significance of the changes (i.e. switches from randomness to non-randomness) in the seismic process, by means of a simultaneous (multivariate) analysis of the so called increments of cumulative time, distances and energy (ICT, ICD and ICE) between consecutive earthquakes in the catalog.

The manuscript illustrates the results obtained by the method's application to real data, from the Southern California earthquake catalog, as well as to synthetic sequences of events, generated by two different non-linear models, one non-Earth specific (i.e. the Lorenz model for an incompressible fluid) and one more closely connected to earthquakes (i.e. the Crack Fusion model).

I believe the proposed approach is interesting and pretty general, and the method might be further expanded to explore other possible patterns in earthquakes occurrence. Therefore in my opinion the manuscript is definitely worth publication, after some necessary revision, taking into account following comments and suggestions to the authors.

**General comments:**

1) Language, as well as text organization, should be significantly improved throughout the manuscript. Specifically, the organization of the manuscript in sections and sub-sections could be improved as follows:

- Used data and methods would be better split in two different sections. Data description (i.e. text from line 100 and up to line 120) could be slightly expanded (see specific comments below);
- Method description (i.e. text from line 121 to line 201) should be included in a separate "Method" section;
- Method testing on synthetic data, generated by Lorenz and Crack-fusion models, should be included in a separate section. The section should include text from line 202 to 250, plus lines from 264 to 290 (including current figures 4 and 5) and could be titled "Testing the method on models";
- The section "Results and discussion" should be focused on results obtained from real data only;
- The part of the text describing the analysis performed for different representative magnitude thresholds, namely M3.6 and M4.6, from line 468 to line 512, should be included in a separate sub-section (possible title: "Testing stability of results with respect to minimum magnitude").

Careful proofreading would make the text more readable and understandable in several parts: some terms seem not to be used properly, some parts are unnecessarily repeated and some quite obvious statements could be removed (see also specific comments below).

2) The method description is quite general. I would suggest the authors to add, if possible, an Appendix explicitly explaining how the method is applied to real earthquake data, namely:

- How the "derivative quantities" ICT(i), ICD(i) and ICE(i) are computed and how they are "normed" to their standard deviation (i.e. which data are used to compute the standard deviation)? How it looks like the distribution of these quantities (e.g. is it Gaussian)?

- How $D^2$ is computed from the three quantities ICT(i), ICD(i) and ICE(i)? Which equation is used? How it looks like the covariance matrix Si (see lines 174-175) in terms of ICT, ICD and ICE?

- How it is estimated the number of degrees of freedom for the specific F-test?

These might appear obvious aspects; however in my opinion such detailed information would help the reader to better understand the method, and would make the obtained results replicable.

3) A critical point in the analysis could be related to the duration of the temporal windows associated with a fixed number of events. In fact, considering a fixed number of events (i.e. n=50 events), the related time span is longer during periods of low activity ("quiet" periods), while it is shorter during periods of high activity, such as during aftershock sequences (particularly after large earthquakes). The variability of the temporal window might have some influence on the obtained results. This is suggested also by the results presented in figure 17, where a smaller number of events is included in the analysis; in this case it is shown that the smoothing for n=50 does not allow to appreciate the low value of MD preceding some large events, while for n=30 (corresponding to a shorter time span) the pattern is visible again. In my opinion it could be interesting to perform a similar analysis over time windows of fixed length. I would suggest the authors to add a comment about this aspect.

4) Based on results illustrated in figures 13-15, the authors conclude that during periods of relatively small earthquakes, with magnitudes not exceeding M=4.6 and far from strong events, seismic activity is close to random. On the other side, after larger events a switch from random to more regular behavior is detected. Thus it seems that a different behavior is detected depending on magnitude. The specific magnitude threshold M4.6 considered in this study has been proposed by Hough (1997). Would this "critical" magnitude threshold depend on the considered region? How this threshold could be determined for other areas? I feel authors should expand a bit the comments about this difference between relatively "strong" and "small" earthquakes.

**Specific comments:**

- Consider replacing "exactly" by "specifically" throughout the text.
- Abstract (lines 19-20): the sentence should be reformulated; it is not clear that the " *different representative threshold values*" refer to the completeness magnitude threshold of the data.
- Consider replacing "normed" by "normalized" in data description.
- Introduction (lines 71-73). The sentence *"In common parlance...replaced by disorder."* is not clear. It should be reformulated or removed.
- Used data (lines 117-120): Data completeness for M2.6 and above is assumed by the authors, relying on earlier studies and analyses. However the meaning of

the sentence: *"we declare that take responsibility on the trustworthy of our analysis"* is not clear. Data description could be expanded, e.g. showing the frequency-magnitude distribution of events, commenting on the number of large events, etc.

- Method (lines 157-159): The sentence *"To be more precise... of the investigated process."* does not seem to add any information about the relevance of the considered data sets. The sentence should be removed or reformulated, so as to explain why the considered features are adequate to this specific analysis.
- The definition of "three dimensional system" and the related use of abbreviations (e.g. 3D) should be used consistently throughout the text.
- Figure 3. Top panel: authors may wish to consider providing seismic energy in logarithmic scale, if appropriate. Bottom panel: the meaning of white circles and bars should be explained in caption.
- Results from real seismicity (lines 300-306). I would suggest the authors to consider the recent paper by Kossobokov and Nekrasova (2017 - "Characterizing Aftershock Sequences of the Recent Strong Earthquakes in Central Italy", Pure Appl. Geophys.. 174: 3713–3723), and include the related reference in their comments, if they feel appropriate.
- Figures 7-9 are similar and can be grouped into a single figure (panels a, b, c). The caption would be the same, except for the time span to be provided for each of the three panels This should also facilitate the comparison and avoid repetitions in the text.
- Results (lines 362-366). The sentence *"Here need to be underlined that... becoming again dominant with respect to the rate of aftershocks' occurrence [Godano, C., Tramelli, 2016]."* is not clear and should be reformulated.
- Results (lines 386-387) Consider replacing *"assessing it by the variation of ICT(i), ICD(i) and ICE(i) data."* by: "assessing it by MD variation." Similarly, at lines 395-396, replace: *"according to distribution of its ICT(i), ICD(i) and ICE(i) characteristics"* by: "according to MD values".
- Results (lines 400-402). The sentence *"It can be added here...(e.g. M6.0)"* is not clear. It should be reformulated in a more specific way, avoiding statements like "*strong, but not strongest*".
- In figure 13 the line of MD threshold is missing. In figures from 6 to 17, the title of axis should be the same (e.g. "n (first event in window)"; the information in brackets can be eventually given in the caption).
- Results: line 426. The sequential number for M5.12 earthquake is missing.
- Testing stability of results with respect to minimum magnitude threshold (line 473). As in the abstract, I feel it should be explained the meaning of "higher representative threshold values".  In fact, magnitude is not clearly mentioned in this paragraph, and the completeness magnitude (i.e. the representative threshold) has been assumed to be M2.6 in the section data description.
- Discussion (lines 523-525). The authors state that *"The period, for which such deviation from the random behavior can last, depends on the amount of seismic energy released by the strong earthquake."* Was this correlation analyzed formally? Did the authors check whether the duration of periods of non-random behavior actually correlates with the seismic energy released by the strong earthquakes? It would be interesting to see how such durations compare with the time windows widely considered for aftershocks identification.

---

## Author Comment (AC2) · 21 May 2019

See enclosed PDF file for detailed comments.
Please also note the supplement to this comment:
https://www.nonlin-processes-geophys-discuss.net/npg-2018-57/npg-2018-57-RC2supplement.pdf

**Comments on**
**"Mahalanobis distance based recognition of changes in the dynamics of seismic process"**
by Teimuraz Matcharashvili, Zbigniew Czechowski, Natalia Zhukova

A novel method is proposed, based on well-established Mahalanobis metric, to quantitatively assess significance of the changes (i.e. switches from randomness to nonrandomness) in the seismic process, by means of a simultaneous (multivariate) analysis of the so called increments of cumulative time, distances and energy (ICT, ICD and ICE) between consecutive earthquakes in the catalog. The manuscript illustrates the results obtained by the method's application to real data, from the Southern California earthquake catalog, as well as to synthetic sequences of events, generated by two different non-linear models, one non-Earth specific (i.e. the Lorenz model for an incompressible fluid) and one more closely connected to earthquakes (i.e. the Crack Fusion model). I believe the proposed approach is interesting and pretty general, and the method might be further expanded to explore other possible patterns in earthquakes occurrence. Therefore in my opinion the manuscript is definitely worth publication, after some necessary revision, taking into account following comments and suggestions to the authors.

**General comments:**

**1**) Language, as well as text organization, should be significantly improved throughout the manuscript.

We did our best to improve language of our manuscript.

Specifically, the organization of the manuscript in sections and sub-sections could be improved as follows:
- Used data and methods would be better split in two different sections.
  Data description (i.e. text from line 100 and up to line 120) could be slightly expanded

(see specific comments below);

Done

- Method description (i.e. text from line 121 to line 201) should be included in a separate "Method" section;

Done

- Method testing on synthetic data, generated by Lorenz and Crack-fusion models, should be included in a separate section. The section should include text from line 202 to 250, plus lines from 264 to 290 (including current figures 4 and 5) and could be titled "Testing the method on models";

Done

- The section "Results and discussion" should be focused on results obtained from real data only;

Done

- The part of the text describing the analysis performed for different representative magnitude thresholds, namely M3.6 and M4.6, from line 468 to line 512, should be included in a separate sub-section (possible title: "Testing stability of results with respect to minimum magnitude").

We are especially grateful to reviewer for suggestion of this subsection.

Careful proofreading would make the text more readable and understandable in several parts: some terms seem not to be used properly, some parts are unnecessarily repeated and some quite obvious statements could be removed (see also specific comments below).

Thanks for these remarks, we took them into account in our revision.

**2**) The method description is quite general. I would suggest the authors to add, if possible, an Appendix explicitly explaining how the method is applied to real earthquake data, namely: - How the "derivative quantities" ICT(i), ICD(i) and ICE(i) are computed and how they are "normed" to their standard deviation (i.e. which data are used to compute the standard deviation)?

In order to explain how we calculated the derivative quantities we add a short explanation to the revised version but we think that there is no need to put it in a separate Appendix.

We start from the first earthquake from the catalogue (for the focused time period from 1975 to 2017), which we consider as a starting point and follow accordingly to the time sequence. Therefore, ICT(i) is the i-th interevent time (i.e. time between i-th earthquake and (i-1)-th earthquake; ICD(i) is the distance between the consecutive events and ICE(i) is the energy of the i-th earthquake. On the other hand, we can also define the quantities in terms of increments of the cumulative sums, i.e., ICT(i), ICD(i) and ICE(i) are increments of cumulative sums of: interevent times, interevent distances and seismic energy released by consecutive earthquakes, respectively.

Regarding to the normalization procedure standard deviations were calculated for each of ICT(i), ICD(i) and ICE(i) data set and then the data sets were normalized to have its standard deviations equal to one.

How it looks like the distribution of these quantities (e.g. is it Gaussian)?

We are grateful to Dr. Peresan for important question which in fact remains not finally resolved at this time. We mean the question about dynamics of seismic process what directly is connected to the distribution functions describing it in  different domains. The question needs special and careful consideration. Of course there are some accepted models approximately describing the entire process but if we  accept the fact that dynamical features of complex seismic process is changing in space and time then we should agree that sometimes the process will be better described by bell-like shape distribution while in other cases features of process will be  better represented in terms of long tailed distribution. In this research we did not have intention to provide a deep analysis of this complicated question (though definitely we plan to come back to this question later). Of course we can present a histogram of the distribution but it is not used further in our work.

- How D2 is computed from the three quantities ICT(i), ICD(i) and ICE(i)? Which equation is used? How it looks like the covariance matrix Si (see lines 174-175) in terms of ICT, ICD and ICE?

In order to explain better the method we replaced a part of the text from section "Methods and analysis" by the new one with additional formulas. We believe that now the revised section is clear.

- How it is estimated the number of degrees of freedom for the specific F-test?
These might appear obvious aspects; however in my opinion such detailed information would help the reader to better understand the method, and would make the obtained results replicable.

Regarding the number of degrees of freedom see e.g. McLachlan, G. J. (1999), Mahalanobis distance. Resonance, 6, 20–26.; Sinclair, T. International Journal of Forecasting 29 (2013) 736–750). Here p-is dimension of data or number of compared columns in each group, which in our

case was 3. Second degree of freedom was calculated as n1+n2-p-1 where, as it was mentioned above, n1 and n2 are the number of samples in each compared groups (50 in our case).

**3**) A critical point in the analysis could be related to the duration of the temporal windows associated with a fixed number of events. In fact, considering a fixed number of events (i.e. n=50 events), the related time span is longer during periods of low activity ("quiet" periods), while it is shorter during periods of high activity, such as during aftershock sequences (particularly after large earthquakes). The variability of the temporal window might have some influence on the obtained results. This is suggested also by the results presented in figure 17, where a smaller number of events is included in the analysis; in this case it is shown that the smoothing for n=50 does not allow to appreciate the low value of MD preceding some large events, while for n=30 (corresponding to a shorter time span) the pattern is visible again. In my opinion it could be interesting to perform a similar analysis over time windows of fixed length. I would suggest the authors to add a comment about this aspect.

We definitely agree with the reviewer. Such analyses, like that presented in this manuscript logically presume testing by means of use of both fixed length data windows as well as fixed time duration windows. In this research we started from the analysis on the sliding windows with fixed number of data. Obtained results are interesting and shows that that there is a necessity to continue work for the case when windows are of fixed time duration. Thus we agree that fixed time window analysis is a next necessary step. At this time we are trying to resolve several calculation problems related with analysis with windows with fixed time duration e.g. question of comparison of groups with different number of events.

**4**) Based on results illustrated in figures 13-15, the authors conclude that during periods of relatively small earthquakes, with magnitudes not exceeding M=4.6 and far from strong events, seismic activity is close to random. On the other side, after larger events a switch from random to more regular behavior is detected. Thus it seems that a different behavior is detected depending on magnitude. The specific magnitude threshold M4.6 considered in this study has been proposed by Hough (1997). Would this "critical" magnitude threshold depend on the considered region? How this threshold could be determined for other areas? I feel authors should expand a bit the comments about this difference between relatively "strong" and "small" earthquakes.

We described results of our analysis presented in Figs. 13-15 basing on the contemporary vision of seismic process. Magnitude threshold was selected according to Hough (1997) but as we see, if we will change the threshold value, results will not be changed considerably. Main finding in this and previous our researches, in our opinion, is that extent of regularity in the seismic process is changing from 'close to regular' to 'closer to random' behavior. What we really see is the relation between a size of earthquake and duration of period with non-random behavior. Of course it is not clear at this moment what is the exact character of this relation but for three main earthquakes in SC catalogue a presence of such relationship seems to be obvious. At the same time these results are preliminary and of course it is necessary to continue analysis on this catalogue as well as on other catalogues. In this respect we hope to find collaborator colleagues from different countries which will agree to participate in our research.

**Specific comments:**

• Consider replacing "exactly" by "specifically" throughout the text.
  According to reviewer's suggestion in most cases we replaced "exactly" by "specifically".

• Abstract (lines 19-20): the sentence should be reformulated; it is not clear that the " *different representative threshold values*" refer to the completeness magnitude threshold of the data.

      In revised version this sentence now reads as: "Analysis of variability in the extent of regularity of seismic process has been accomplished for different completeness magnitude threshold."

• Consider replacing "normed" by "normalized" in data description.
Done

• Introduction (lines 71-73). The sentence *"In common parlance...replaced by disorder."* is not clear. It should be reformulated or removed.

We accept  reviewers suggestion and removed this sentence from the revised version.

• Used data (lines 117-120): Data completeness for M2.6 and above is assumed by the authors, relying on earlier studies and analyses. However the meaning of the sentence: *"we declare that take responsibility on the trustworthy of our analysis"* is not clear. Data description could be expanded, e.g. showing the frequency-magnitude distribution of events, commenting on the number of large events, etc.

Mentioned sentence is removed from revised manuscript.

Regarding frequency-magnitude distribution of events. We do not want to speak in our manuscript about well known details on the completeness of SC catalogue. At the same time in accordance with reviewers request we show here a figure with the Gutenberg–Richter relationship indicating that magnitude of completeness is M=2.6 for used Southern Californian earthquake catalogue.

[Figure]

• Method (lines 157-159): The sentence *"To be more precise... of the investigated process."* does not seem to add any information about the relevance of the considered data sets. The sentence should be removed or reformulated, so as to explain why the considered features are adequate to this specific analysis.

Here we intended to point the importance of selection of data sets which analysis will give correct understanding of the target process. Our experience as readers and reviewers warns us that it happens not  rarely when authors base their results on data sets which  are not directly related to the process of interest. According to the reviewer suggestion we remove the sentence ("*To be more precise... of the investigated process.*") from the revised version and instead add a new sentence: "For this, in order to have data sets of similar physical sense enabling to assess dynamical features of seismicity in its three domains, as was mentioned above, we selected *ICT(i), ICD(i)* and *ICE(i)* data sets."

• The definition of "three dimensional system" and the related use of abbreviations (e.g. 3D) should be used consistently throughout the text.

Indeed "three dimensional system" is mentioned just one time and after throughout the text we always use abbreviations - 3D.

• Figure 3. Top panel: authors may wish to consider providing seismic energy in logarithmic scale, if appropriate. Bottom panel: the meaning of white circles and bars should be explained in caption.

In our opinion it is better to leave upper plot in Fig.3 in present form. As for bottom plot here we show averages of MD values (white circles) and corresponding standard deviations (grey error bars) calculated for each consecutive windows comparing original and each out of 100 randomized catalogues.

• Results from real seismicity (lines 300-306). I would suggest the authors to consider the recent paper by Kossobokov and Nekrasova (2017 -"Characterizing Aftershock Sequences of the Recent Strong Earthquakes in Central Italy", Pure Appl. Geophys.. 174: 3713–3723), and include the related reference in their comments, if they feel appropriate.

Mentioned article is very interesting and in general support our vision on variable dynamics of seismic process it is cited in revised version.

• Figures 7-9 are similar and can be grouped into a single figure (panels a, b, c). The caption would be the same, except for the time span to be provided for each of the three panels This should also facilitate the comparison and avoid repetitions in the text.

We are grateful for this remark to reviewer, but it seems to us that in the present form figures with separate captions would be more comfortable for readers to understand what was done.

• Results (lines 362-366). The sentence *"Here need to be underlined that...becoming again dominant with respect to the rate of aftershocks' occurrence [Godano, C., Tramelli, 2016]."* is not clear and should be reformulated. *Godano, C., Tramelli, 2016*

We removed a part of the sentence, i.e., "*becoming again dominant with respect to the rate of aftershocks' occurrence*" in the revised version.

• Results (lines 386-387) Consider replacing "*assessing it by the variation of ICT(i), ICD(i) and ICE(i) data.*" by: "assessing it by MD variation." Similarly, at lines 395-396, replace: *"according to distribution of its ICT(i), ICD(i) and ICE(i) characteristics*" by: "according to MD values".

We gratefully accept reviewers remark. Now these parts of manuscript reads as: "Most important still is the fact that prior to almost all strong earthquakes, in periods which can be regarded as relatively calm, the original seismic process is indistinguishable from random process, assessing it by the variation of MD values calculated for windows of 50 data sequences of *ICT(i), ICD(i)* and *ICE(i)* characteristics." And: "As follows from these figures there are no windows in which original seismic process, according to MD values calculated for windows of *ICT(i), ICD(i)* and *ICE(i)* characteristics, can be regarded as random-like."

• Results (lines 400-402). The sentence "*It can be added here...(e.g. M6.0)"* is not clear. It should be reformulated in a more specific way, avoiding statements like "*strong, but not strongest*".

Now the sentence reads as: "It can be added here that the similar was situation for the sequences of small events occurred also after other strong earthquakes in the analyzed catalogue."

• In figure 13 the line of MD threshold is missing. In figures from 6 to 17, the title of axis should be the same (e.g. "n (first event in window)"; the information in brackets can be eventually given in the caption).

Done

• Results: line 426. The sequential number for M5.12 earthquake is missing.

Done

• Testing stability of results with respect to minimum magnitude threshold (line 473). As in the abstract, I feel it should be explained the meaning of "higher representative threshold values". In fact, magnitude is not clearly mentioned in this paragraph, and the completeness magnitude (i.e. the representative threshold) has been assumed to be M2.6 in the section data description.

Done

• Discussion (lines 523-525). The authors state that *"The period, for which such deviation from the random behavior can last, depends on the amount of seismic energy released by the strong earthquake*." Was this correlation analyzed formally? Did the authors check whether the duration of periods of non-random behavior actually correlates with the seismic energy released by the strong earthquakes? It would be interesting to see how such durations compare with the time windows widely considered for aftershocks identification.

Here we just base our conclusion on the results we get from our analysis and which do not contradicts to accepted views on the time duration of aftershock activity initiated by strong earthquakes. So no additional testing has been carried out.

---

## Author Response (AR2)

Answers of authors of manuscript npg-2018-57, to reviewers.

We again express gratitude to both reviewers who noted about necessity of language correction of our manuscript.
Present version of our manuscript has been professionally proofread by Proof-Reading-Service.com Ltd, Devonshire, UK.

As for suggestions of E. Papadimitriou:
-The Map in Fig. 1 has already been changed during first revision. In present version we decide not to change map because in this case part of EQs in considered catalogue will not be shown in the map (Fig.1).

-Also during first revision we explained why we prefer to use seismic energy but not moment in this work. Moreover, we do not see reason to show seismic moment in Figure 5 while through the entire manuscript we speak about seismic energy.

Other suggestions were gratefully accepted and figures 5 and 6 are now corrected.

We also note that we added footnote in page 16 and one sentence in the summary where we say that recent strong earthquakes in California we consider as confirmation of pointed in the manuscript idea, that after long series of randomlike windows may follow window with strong earthquakes.